



# Multi-scale analysis and Modeling of aeromagnetic data over the Bétaré-Oya area in the Eastern Cameroon, for structural evidences investigations.

**Christian Emile Nyaban[a]; Théophile Ndougsa-Mbarga[a, b*]; Marcelin Bikoro-Bi-Alou[c];**

**Stella Amina Manekeng Tadjouteu[a]; Stephane Patrick Assembe[a, d]**

[a]Postgraduate School of Sciences, Technologies & Geosciences, University of Yaoundé I, Yaoundé, Cameroon.

[b]Department of Physics, Advanced Teachers' Training College, University of Yaoundé I, P.O. Box 47 Yaoundé Cameroon.

[c]Department of Earth Sciences, Faculty of Science, University of Maroua, Maroua, Cameroon.

[d]Department of Physics, Faculty of Science, University of Bamenda, Bamenda, Cameroon.

Correspondence should be addressed to Ndougsa Mbarga Théophile; [*]tndougsa@yahoo.fr

**ABSTRACT:**

This study was carried out in the Lom series in Cameroun, at the border with Central African Republic located between the latitudes 5º30'-6ºN and the longitudes 13º30'-14º45'E. A multi-scale analysis of aeromagnetic data combining tilt derivative, Euler deconvolution, upward continuation and the 2.75D modelling was used. The following conclusion were drawn: 1- Several major families of faults were mapped. Their orientations are ENE-WSW, E-W, NW-SE, N-S with a NE-SW prevalence. The latter are predominantly sub-vertical with NW and SW dips and appear to be prospective for the future mining investigation. 2-The evidence of compression, folding and shearing axis, was concluded from superposition of null contours of the tilt-derivative and Euler deconvolution. The evidence of the local tectonics principally due to several deformation episodes (D1, D2 and D4) associated with NE-SW, E-W, and NW-SE events respectively. 3- Depths of interpreted faults ranges from 1000 to 3400 m. 4- Several linear structures correlating with known mylonitic veins were identified. These are associated with the Lom faults and represent the contacts between the Lom series and the granito-gneissic rocks; we concluded the intense foldings caused by senestral and dextral NE-SW and NW-SE





stumps; 5- We propose a structural model of the top of the crust (schists, gneisses, granites) that
delineates principal intrusions (porphyroid granite, garnet gneiss, syenites, micaschists,
Graphite and Garnet gneiss) responsible for the observed anomalies. The 2.75D modelling
revealed; many faults with a depth greater than 1200 m and confirmed the observations from
RTE-TMI, Tilt derivative and Euler deconvolution; 6- We developed lithologic profile of
Betare Oya basin.
**Keywords: Aeromagnetic data, multi-scale analysis, 2.75D modelling, faults.**
**1. Introduction**
Magnetic method has a renewed interest for solid mineral, hydrocarbons, and geological
researches. During data interpretation, the first crucial step is the removal of the effect of deep-
seated structures from the observed total magnetic field to enhance shallow body signatures
(Ndougsa et al., 2007). These shallow bodies are generally associated to magnetic minerals
such as magnetite, hematite which are contained by iron ore deposit (Ndougsa et al., 2013). The
second step is mapping causative body's edges, which is fundamental to the use of potential
field data for geological mapping. The edge detection techniques are used to distinguish
between different sizes and different depths of the geological discontinuities (Oruç et al., 2011).
In the last few years, there have been several methods proposed to help normalizing the
magnetic signatures in images. Cordell and Grauch, (1985) have suggested a method to locate
horizontal extents of the sources from the maxima of horizontal gradient of the pseudo-gravity
computed from the magnetic anomalies. Verduzco et al., (2004) developed tilt derivative from
gravity or magnetic field anomaly maps using the horizontal gradient magnitude of the tilt
derivative as an edge detector.
Salem et al., (2008) developed a new interpretation method for gridded magnetic data based
on the tilt derivative, without specifying prior information about the nature of the source. In this
article, we suggest another approach which consists in the location of vertical contacts by using
the maxima and horizontal edge of tilt derivative.






## 2. Geological and tectonic setting

### 2.1. Regional setting.

The following structural domains can be distinguished in the Pan-African belt north of the
Congo craton (Toteu et al., 2004; Fig. 1.A):

(a) A pre-collisional stage that includes the emplacement of pre-tectonic calc-alkaline
granitoids (e.g., at 660‑670 Ma);

(b) A syn-collisional stage inducing crustal thickening and delamination of the subcrustal
lithospheric mantle and comprising D1 and D2 deformations and S-type granitoids
(640‑610 Ma; Toteu et al., 2004);

(c) A post-collisional stage associated with D3 deformation (nappe and wrench)
concomitant with exhumation of granulites, development of D4 shear zones, and
emplacement of late-tectonic calc-alkaline to sub-alkaline granitoids (600‑570 Ma).

The Pan-African formations of Cameroon belong to the mobile zone of Central Africa (Bessoles
et al.,1980), also known as the Oubanguide chain (Poidevin, 1985). It is attached to the East to
PanAfrican formations of the Mozambican belt of submeridian orientation. To the West, it
extends to the North of Brazil by the Sergipe range. Two large dextral mylonitic shear zones,
the Sanaga Fault (Dumont, 1986) and the Cameroon Center Shear Zone, cross Cameroon from
northeast to southwest. These major shears belong to the Oubanguid setback zone (Rolin, 1995),
which continually follows from the Gulf of Guinea to the Gulf of Aden (Cornacchia et al.,
1983). Geologically, the Pan-African mobile chain is composed of granites, schists,
micaschists, and migmatites (Poidevin, 1985).









**Figure 1.A** Geologic map of Cameroon, showing major lithotectonic units: ASZ, Adamaoua
shear zone; CCSZ, Central Cameroon shear zone; TBSZ, Tcholliré- Banyo shear zone,



modified from Kankeu et al. (2009) as a document available in a public domain. The location
of the study area is marked by a box and shown in detail in Figure 1B.
**2.2. Local setting.**
The study area is located in eastern Cameroon; it is bounded by north latitudes 5º30'-6º, and
east longitudes 13º30'-14º45' (Fig. 1.B). The lithology comprises the Lom series constituted of
Neoproterozoic rocks sequence consisting of metasedimentary and metavolcanic rocks with
late granitic intrusions (Ngako et al., 2003). The lithologic units have a strong NE-SW regional
foliation deflected in places by the granitic pluton reflecting dextral and sinistral shear senses.
The rocks have been metamorphosed to greenschist facies and hydrothermal alteration
especially around the granitic plutons (Odey Omang et al., 2014). Gold is sporadically identified
in NE-SW quartz veins associated with early pyrite whereas a vug-filling late pyritization event
is barren (Asaah, 2010; Nih Fon et al., 2012).

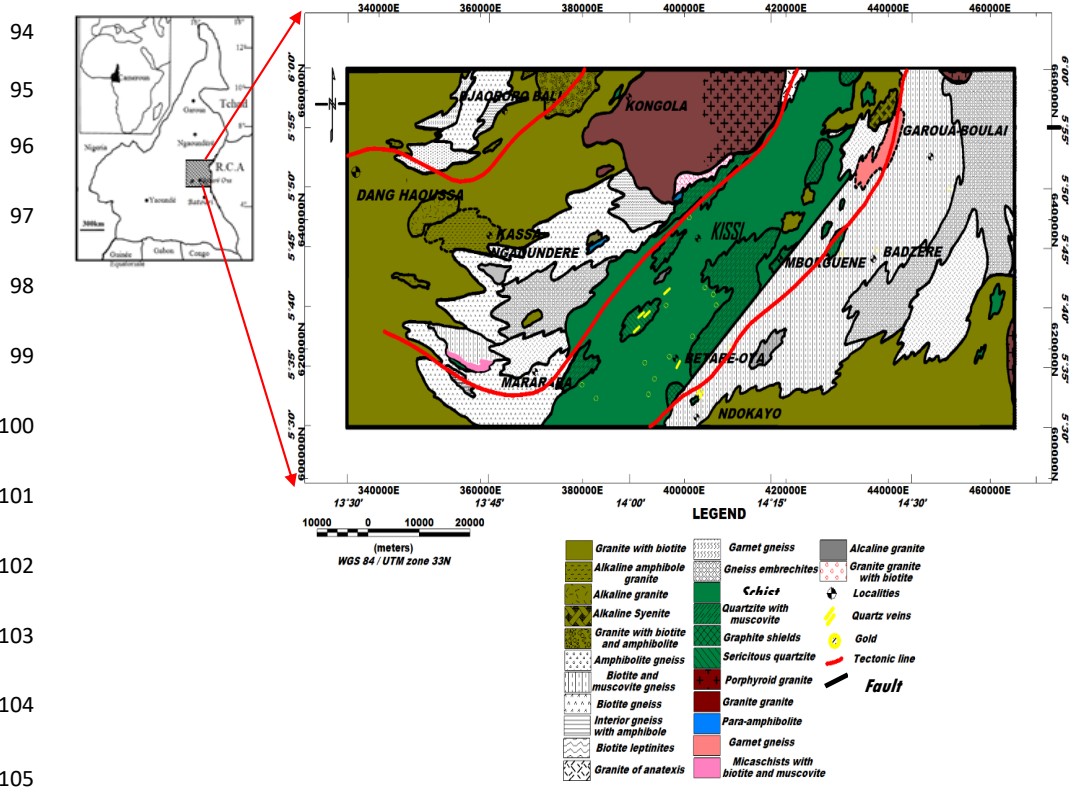





**Figure 1.B** Geological map of the study area (Gazel and Gerard, 1954 modified as a document available in a public domain). In the center we have the Lom series marked by its greenschist facies. We can also perceive in red the tectonic lines that cross the study area.

The orography and hydrographic network would be structurally guided (Kouske, 2006), subdivided into three major morphological units. The high altitude unit (800-1092 m) which is a vast peneplain enameled by interfluves with multiple vertices of alignment oriented NW-SE; N-S and NE-SW; The low altitude unit (652-760 m) which is a large flat-bottomed depression, in the center of which is a U-shaped valley, oriented NE-SW within which flows the Lom river and the intermediate unit (760-860 m) which corresponds to a long NE-SW oriented cliff connecting the high altitude unit to that of low altitude. In its northern part, this unit has an E-W orientation.

### 2.3. Geophysical constraints

Seismic anisotropy in Cameroon has been studied by Koch et al., (2012) through analysis of SKS splitting allows to identify four regions of distinct anisotropy: moderately strong NE-SW oriented fast polarization directions ($\delta t \approx 1.0$ s) beneath two region: the Congo Craton in the south and the Garoua rift in the north; weak anisotropy ($\delta t \approx 0.3$ s) between the Congo Craton and the CVL; N-S oriented fast polarization directions within the CVL, with $\delta t \approx 0.7$ s. (Koch et al., 2012). Jean Benkhelil et al., (2002) used seismic data and proposed structural and chronostratigraphic scheme of the southern Cameroon basin (clayey sand, dolomitic to calcitic sandstone, marls and sandstone, dolomitic sandstone, granite, gneiss).

Gravity studies are carried out, Tadjou et al., (2004) identify many structures like contacts, dykes, fractures, and faults in the transition zone between the Congo Craton and the Pan-African Belt in Central Africa. Shandini et al., (2011) put into evidence in the northern margin of the Congo Craton a deep structure, which corresponds to a classical model of collision suture of the West-African Craton and Pan-African belt.





Owono et al., (2019) used 2.75D modelling of aeromagnetic data in Bertoua and shows
intrusive bodies composed of gneiss and porphyroid granite and some domes with their roof
situated at various depths not exceeding 1800 m from the surface. The structural map of the
study area shows the trending of the structural features observed, namely, NE-SW, NW-SE,
ENE-WSW, and WNW-ESE, respectively, while the E-W and N-S are secondary orientation
of the observed tectonic evidence.
**3. Materials and Methods**
*3.1. Data acquisition and processing.*
The aeromagnetic data were collected in Cameroon by Survair Limited through the
Cameroon/Canada cooperation framework in the 1970s. Data were collected along N-S flight
lines at 750 meters spacing, with a flying height of 235 meters; the measurements involved a
magnetometer with a sensitivity of 0.5 nT (Paterson et al., 1976). Aeromagnetic anomalies map
has been digitized using the geographical information system software (Mapinfo Pro. 16.0) and
interpolated on a 850 m cell-sized grid. The estimate error introduced is 0.28 mm which is
usually considered to be distinctive capacity of human vision (Achilleos, 2010). Gridding and
processing were done with Geosoft v8.4 software. The IGRF-84 reference field values were
removed from the observed magnetic data.
*3.2. Methods*
*3.2.1. Upward continuation theory.*
The upward continuation subjects the observed potential field on a surface, in order to obtain
the field which would be observed on another surface above of the initial surface of observation.
In this study it helps us to easily visualize the effects of the deep sources and to remove the
regional effect. This method was described by (Blakely, 1996).
*3.2.2. The Tilt-angle approach.*
The tilt-angle (Miller and Singh 1994; Verduzco et al., 2004; Salem et al., 2007) is defined
by the equation (1) below for a potential field anomaly T:





$\quad \theta = \tan^{-1} \dfrac{\dfrac{\partial T}{\partial z}}{\dfrac{\partial T}{\partial h}}$ (1) where
$\quad \dfrac{\partial T}{\partial h} = [(\dfrac{\partial T}{\partial x})^2 + (\dfrac{\partial T}{\partial y})^2]^{1/2}$ is the horizontal gradient magnitude and $\dfrac{\partial T}{\partial z}$ is the vertical gradient;
$\quad \dfrac{\partial T}{\partial x}, \dfrac{\partial T}{\partial y}$ are respectively the horizontal gradients along the x and y directions.
In 2007, Salem et al., extended the method to the determination of depth to source by relating
the depth Zc of the source and its horizontal location h to the tilt-angle through equation (2):
$\quad \theta = \tan^{-1}(\dfrac{h}{Zc})$ (2)
This means that the contacts are located for a nil tilt (h = 0) and the depth corresponds to
horizontal distance between 0º and ± 45º contours, i.e. h = ± Zc (Salem et al., 2007).
*3.2.3. Qualitative analysis by Tilt-angle derivative.*
The tilt angle operator can be used for mapping geological structures because it permits to
locate and to delimit their contacts and their shapes (Miller and Singh, 1994). By coupling it to
the extension upward, it becomes more interesting because one obtains the lateral extension of
body but also in depth therefore its three-dimensional shape.
*3.2.4. Euler's Deconvolution.*
This method was introduced by Thompson, (1982) based on the Euler's homogeneity
equation to solve for the source depths for profile data. Reid et al., (1990) extended the operator
to gridded data by using equation (3): w
$\quad \dfrac{(x-x_0)\partial M}{\partial x} + \dfrac{(y-y_0)\partial M}{\partial y} + \dfrac{(z-z_0)\partial M}{\partial z} = N(B-M)$ (3)
where $(x, y, z)$ represent the coordinates of the observation point, $(x_0, y_0, z_0)$ the coordinate of
the magnetic source, $M$ and $B$ are the field at the observation point and regional the field
respectively; and $N$, the structural index, characterizes the variation rate of the field in relation





to the distance due to the type of source (table 1.A). In this study, we take the advantage of the
clustering in depth to define the correct structural index.
**Table 1.A**    Structural index for magnetic sources of different geometries.

| Source | Smellie model | Structural index |
|---|---|---|
| Sphere | Dipole | 3 |
| Vertical line end (pipe) | Pole | 2 |
| Horizontal line (cylinder) | Line of dipoles | 2 |
| Thin bed fault | Line of dipoles | 2 |
| Thin sheet edge | Line poles | 1 |


*3.2.5. 2.75D modeling*.
A very useful variation on the 2D model which removes the restriction of infinite strike
length, and is easier to define than the more complex 3D model, is a model with constant cross-
section extending over a finite strike length. This is known as 2.5D model. When the source
can have different strike extents on either side of the modelled profile, or the strike or plunge
of the body is not perpendicular to the profile, this is called a 2.75D model.
The 2.75D model represents the subsurface as a series of polygonal prisms with horizontal
axes (X) and finite extent in the strike direction (Y). This method was described by Skalbeck et
al., (2005). Geologic models were constructed with GM-SYS operator of Geosoft using the
2.75D modelling algorithm from Won and Bevis (1987), based on the analyses of Rasmussen
and Pedersen (1979). The 2.75D model gives the interpreter control of the third dimension
without the complexity of defining and manipulating a full 3D model.

**4. Results**

After interpolation, data have been reduced to the equator using the Fourier transform
(Inclination I = -11.98 deg, Declination D = -4.96 deg) on January 1, 1970. This transformation





eliminated the tilt of the earth magnetic field due to inclination and positioned anomalies
directly above the corresponding magnetic source.
***4. 1. Interpretation of the aeromagnetic total field reduced to Equator.***
The magnetic field over the Bétaré-Oya area has a complex magnetic pattern (Fig. 2.A). For
better characterization of the geological structures, we subdivided the area into different units:
*Unit A*
The major observable singularity is in the center where a large positive anomaly about 5 km
wide and up to 100 nT is observed. It is oriented NE-SW along the major tectonic feature in
this area, namely the tectonic line of the Sanaga (Fig. 1.A). Comparing with the geological map
in Figure 2, this signal is mainly due to volcano-clastic schists (with gold deposit) also called
Lom schists associated with conglomeratic quartzites with intrusions of granitoids (Kankeu et
al., 2009). Hence, the presence of the anomalies with similar signatures could be related the
circulation of hydrothermal fluids rich in magnetic minerals along the Betaré-Oya Shear Zone
(BOSZ).
*Unit B-C*
In the northeastern part of Bétaré-Oya, particularly around Badzéré, two positive and
heterogeneous bipolar anomalies are observed. The positive pole is slightly more developed
than the negative pole located in the south of the area at Ndokayo, with very long wavelength
of about 22 km. Its amplitude is quite high and reaches 120 nT. It is aligned with the one of the
major foliations in this area trending E-W. The shape and amplitude of these anomalies suggest
high susceptibilities of the causative bodies, such as igneous granitoids know in this area.
*Unit D-E*
In Mararaba and Kassa, there is a large magnetic anomaly with a bipolar shape, whose positive
pole labeled as E in Figure 3 and the negative pole D. It is characterized by a long wavelength
with variable amplitude reaching 150 nT, its approximate direction is ENE-WSW. We can also



222 observe positive and negative anomalies of intensity 100 nT and 20 nT, elongated shapes,

223 circular and semi-circular, short wavelength-oriented ENE-WSW, NW-SE, NE-SW

224 corresponding to structural directions in the study area (Kankeu et al., 2009, Nih Fon et al.,

225 2012).

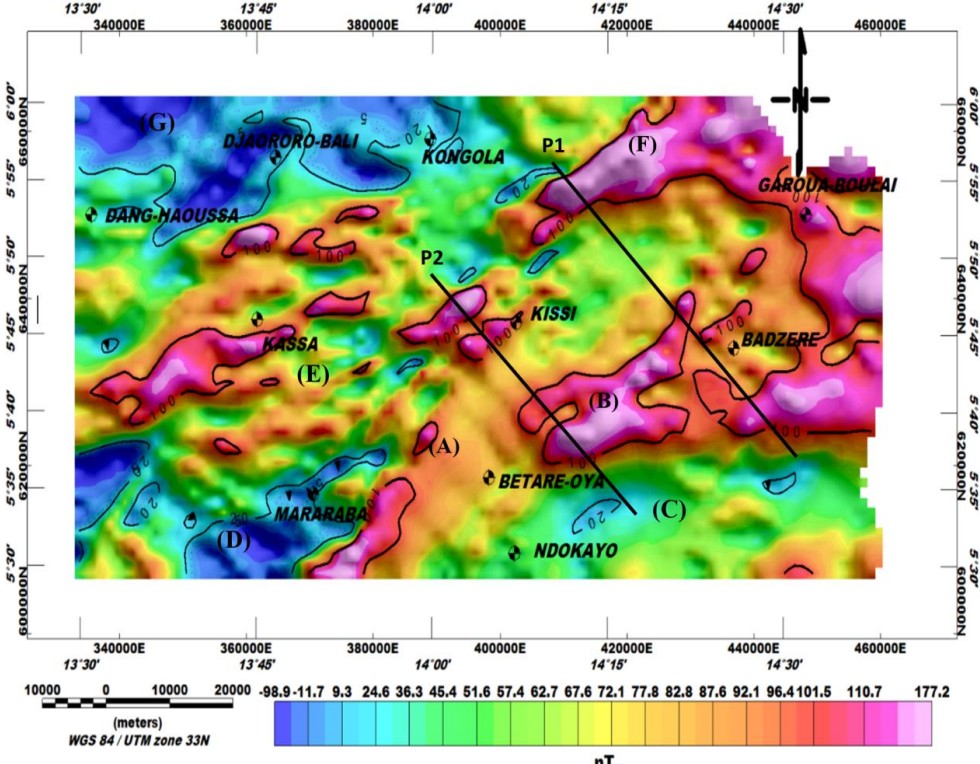

228 **Figure 2.A** Total magnetic intensity (TMI) map reduced to the equator.

229 *Unit F*

230 In the northwestern part of Garoua-Boulai, a positive and heterogeneous anomaly with irregular

231 shapes, normal polarity and a very long wavelength of about 22 km has been observed. Its

232 amplitude is quite high and reaches 177 nT. Its approximate direction is ENE-WSW. It is




probably associated with the meta-volcanic outcrops of the meta-lava within the schistous Lom
series (Regnoult, 1986).
*Unit G*
The lowest magnetic intensities are recorded in the north-west near Djaororo-Bali, where
negative anomalies with amplitudes down to -98.9 nT are found associated with surface meta-
sediments such as modified-biotite gneiss overlying the old metamorphic basement.
**4.  2. Tilt-angle on residual map.**
The residual map is obtained by subtracting the total magnetic field map reduced to the equator
to the regional map. The determination of the optimum regional anomaly map for the study area
lies on the method of Zeng, (1989). This method consists in determining a suitable altitude for
upward continuation in the study area. The extrema of each altitude of upward continuation are
then counted (table1.B). These are points where the gradient is null. Further, a graph of extrema
versus altitudes of upward continuation is plotted (Fig. 2.B). Finally, the suitable altitude (h=10
km) necessary for the upward continuation technique is determined graphically.

**Table 1.B**   Maxima and altitudes of upward continuation.

| Number of maxima | Altitudes of upward continuation (Km) |
|---|---|
| 38 | 0 |
| 19 | 1 |
| 14 | 2 |
| 6 | 3 |
| 5 | 4 |
| 5 | 5 |
| 2 | 6 |
| 1 | 7 |
| 1 | 8 |

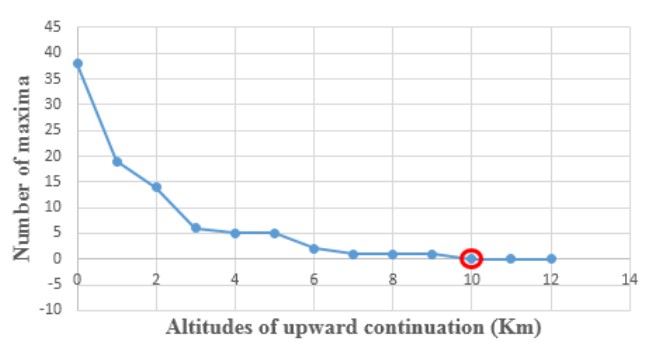

**Figure 2.B**    Number of extrema versus upward continuation height. From h = 10 km (circle in red), the number of maxima becomes constant and does not vary anymore.


| 1 | 9 |
|---|---|
| 0 | 10 |


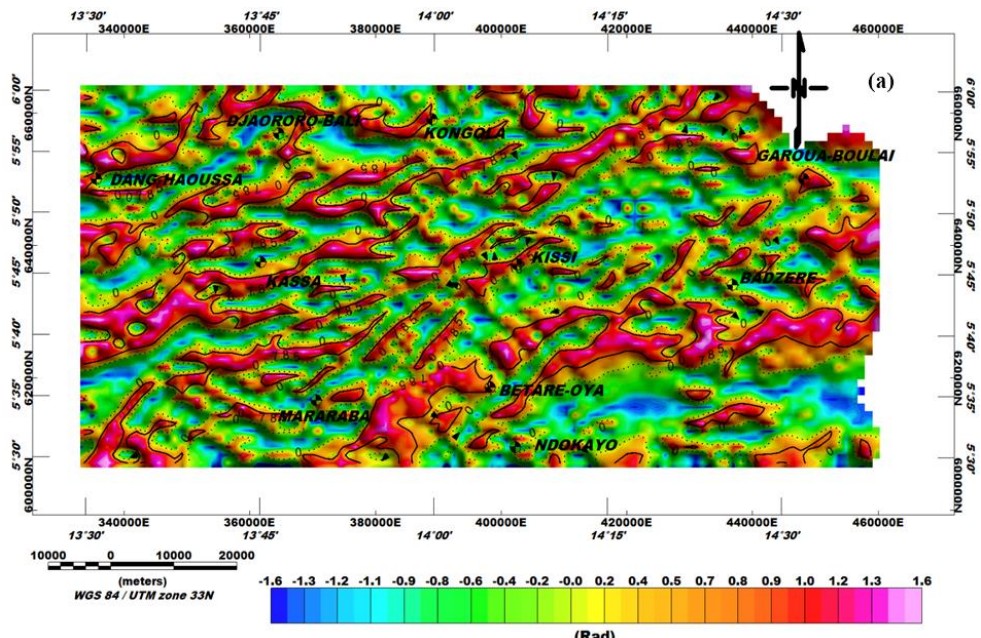


**Figure 2.C**   Tilt angle on residual map.

The generated tilt-angle's map (Fig. 2.C) represents possible lineaments of the study area.

On this map it can clearly be seen that the signal is uniformly distributed in -1,6 rad to 1,6 rad

intervals; thus, making it possible to map the lineaments with a very high resolution. The

presence of several accidents marks the heterogeneity of the basement in this area as well as

the intense deformation undergone by its subsurface. The lineaments and spatial patterns of

geophysical attributes are important information that can be obtained from magnetic

interpretations. Steep features and straight faults are commonly expressed as subtle lineaments

of potential field. This expression can be gradient zones, local anomaly alignments of different

types and shapes, aligned breaks, or discontinuities in the anomaly model.








### *4.3. Structural map.*

To characterize information, we were interested in the peaks of anomalies derived from tilt
angle derivative (Fig. 2.C). We counted 111 lineaments among which: 45 have lengths varying
between (2.5 - 10.8) km; 37 minor lineaments varying between (1.2 - 2.3) km and 29 major
lineaments between (2.4 - 15.6) km. Five structural families NE-SW are observed; ENE-WSW;
E-W; NW-SE; N-S, the major structural direction being NE-SW (Fig. 3.A).

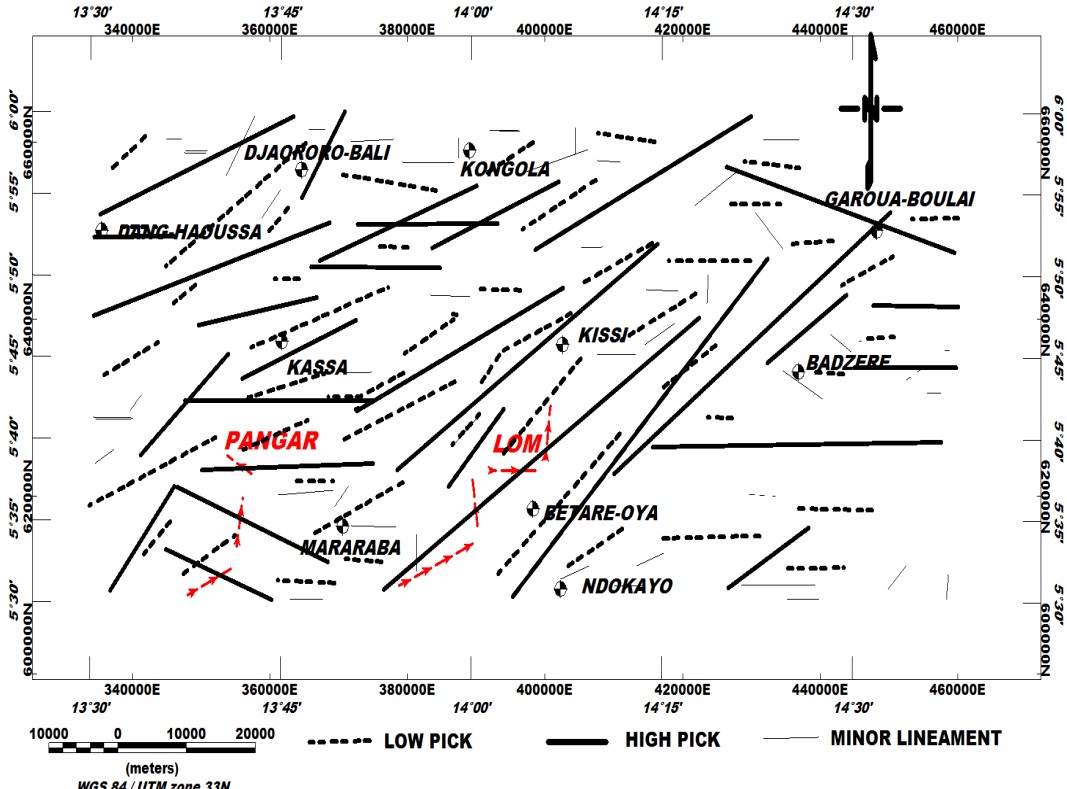


**Figure 3.A**   Structural map of the study area.



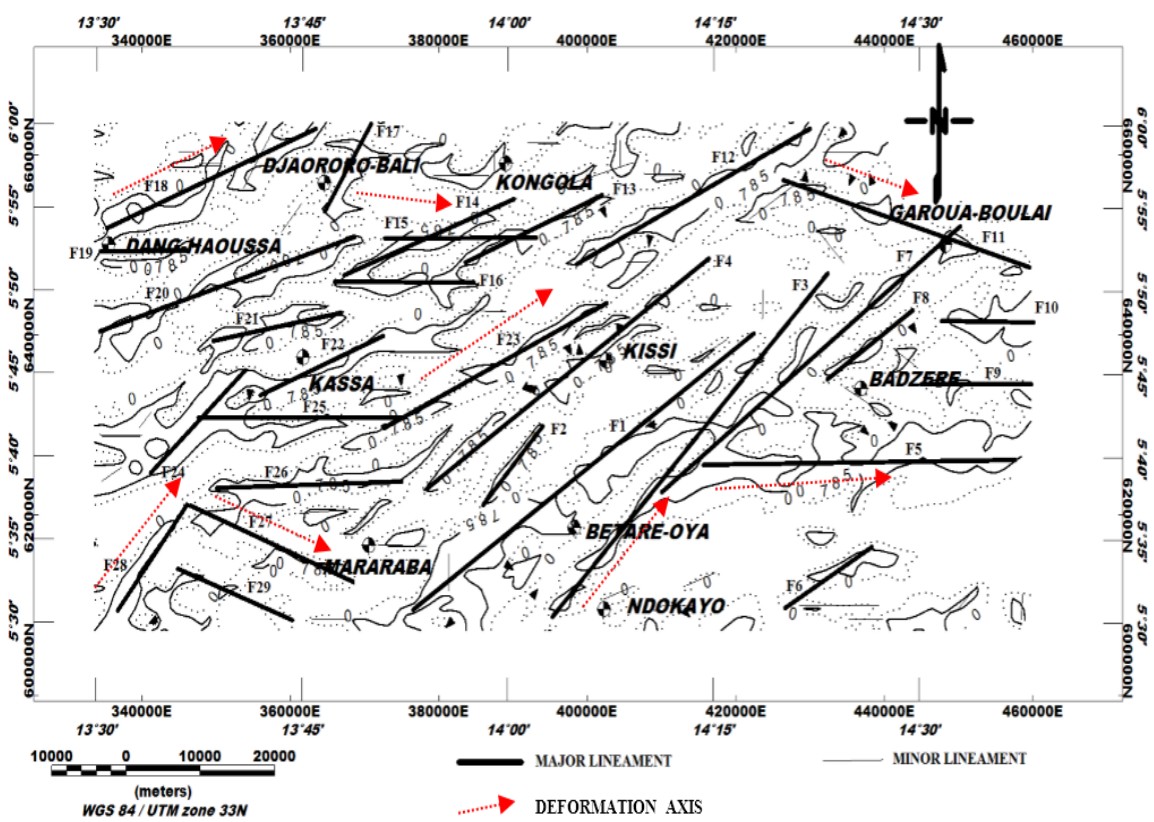

**Figure 3.B**  Major faults map superimposed on tilt-angle contours. On this map we observe

the major regional deformation axes (NE-SW, E-W, ENE-WSW and NW-SE) as well as the

associated faults (F1 to F29).

The longest faults are present at the eastern edge of the Lom series with lengths of more

than 15 km (F1, F3, F7). To the west we also note the NE-SW F4 fault with more than 10 km

length which marks the limit of the Lom series (Fig. 3.B). The most remarkable is the change

of direction of compression or deformation axes. The E-W events marked by the faults F15,

F16, F19, F25, F26 at the eastern edge of the Lom and by the faults F5, F9, F10 in the west,

seem to have been taken up by the tectonic accidents F1, F2, F3, F4, F7, F8, F12, F23

punctuated by the Betaré-oya shear zone (BOSZ). The same phenomenon occurs in the extreme

west of the study area around Dang Haoussa and Mararaba with the ENE-WSW (F13, F14,





F18, F20, F21, F22) and NW-SE (F27, F29) accidents, respectively. These discrepancies
suggest the passage of shear faults. The curvature (type II) structures corresponding to foliations
induce most of the major fault network present in the Bétaré-oya area. In order to confirm the
results obtained by the tilt-derivative, we apply the Euler Deconvolution method.

### *4.4. 3D extension of anomalies.*

By superposing the zero contours of tilt-angle of the residual map, we obtain Figure 3.C

which no perfect superimposition of sources on the previous ones, hence assuming the
heterogeneity of the basement and existence of movements that affected the subsurface
formations. Deep crustal tightening of volcano – clastic rocks in the vicinity of Betaré - Oya
confirms that the site is affected by shear tectonics (Soba, 1989), causing deep and shallow
faults. This is witnessed by the contact between the granito-gneissic rocks and the Lom schists
(Fig. 1.B). These contours delimit the edges of the magnetic source, so their superposition in
depth allows to have an idea about the disposition, the extent, the dip and the shape of the
geological sources responsible for the magnetic anomalies observed.

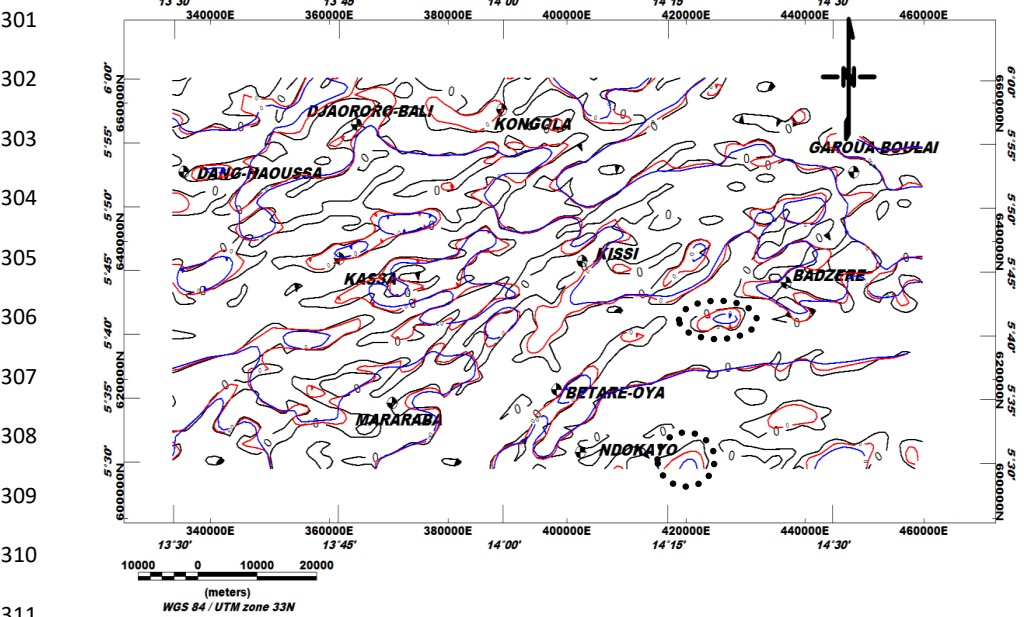





**Figure 3.C**  Superposition of contours (Θ=0º) of Tilt angle of RTE upward continued to 1 km
(red) and 2 km (blue).
East of Ndokayo, Kassa and south-east of Mborguene, several structures lose extension in
depth, taking the form of a basic cone of revolution located on the surface (interrupted circle).
The presence of this regional-scale fold system, which controls all movements in the area
(BOSZ), suggests an interconnection of crustal geological structures by lines of faults and
foliations. Hence the structural elements highlighted in this study (folds, faults, dykes, etc.)
globally belong to Pan-African tectonics.
**4.5. Quantitative analysis**
*4.5.1. Tilt-angle.*
The tilt-angle operator makes it easy to determine the depth of the vertical contacts (Salem
et al., 2007) by estimating the distance between the zero-angle contours and those
corresponding to the values ± 45º (Fig. 3.B). We have determined the average depths interval
ranges from 1 to 3 kilometers for major lineaments (Table 1.C).
*4.5.2. Euler deconvolution.*
Euler's solutions allowed us to verify the position of the contacts obtained by the tilt angle
method as well as their depth.
The superposition of the structural map with Euler's solutions allowed us to delimit deep and
superficial faults, dykes and veins; to delineate tectonic lines established by previous geological
studies (Gazel et al., 1954) and to compare with results from the tilt angle method (Fig. 4.B).






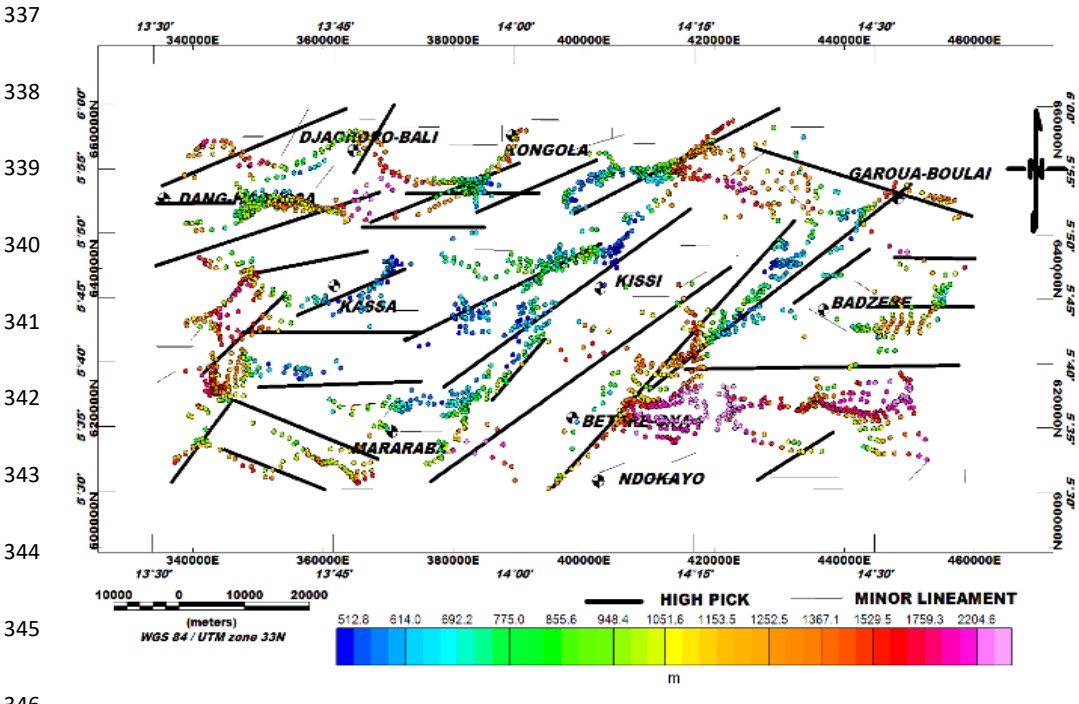

**Figure 4.A**   Euler solution (N=1; W=25; Z=10%).

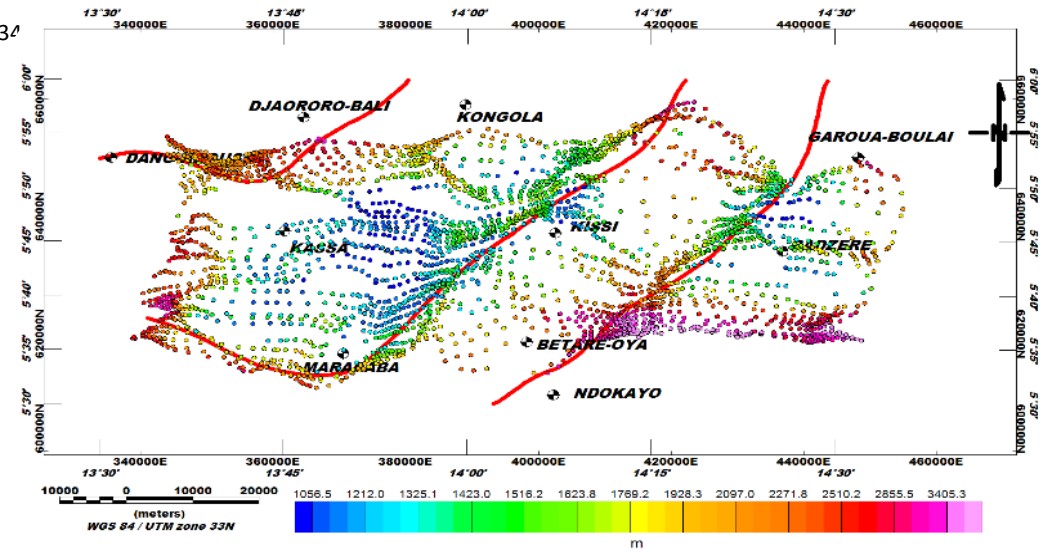

**Figure 4.B**   Euler solution (N=2; W=20; Z=10%). The Euler maps above allow for a

comparative study with the results obtained from the tilt derivative. They also make it possible





to confirm the tectonic lines of the zone (in red) highlighted in the work of Gazel and Gerard,
(1954) and to estimate their depths.
On Euler's solutions map we have perfectly distinguished the limits of the intrusive bodies
and the deeper faults. On these maps, we observe five main directions of structures namely:
NE-SW; ENE-WSW; E-W; NW-SE; N-S (Fig. 3.B). In addition, the vertical contacts are
clearly visible on Euler solutions map and extend over 15 km length.
The deepest accidents are mainly NE-SW to E-W with depths of over 3500 m and are well
located at the eastern limits in the Lom series and the Badzéré gneisses contact zone and also
around the East fault of Bétaré-oya. In the south-west of the map, at Mararaba, Euler's solutions
allow to detect approximately NW-SE faults that was the result of the highlighted tectonic line
(Fig. 4.A) and whose depths are estimated at 3000 m. We obtain depths ranging from 0.5 to 3.6
km. Figure 4.B clearly shows tectonic directions which dominate all subsurface movements of
the study area and their depths ranging from 1 to 3.4 km.
**Table 1.C**    Main faults of Lom series.  This summary table is obtained after comparing the
results from the Euler deconvolution method and the tilt derivative.

| Faults | Directions | Dips | Depths (km) |
|--------|-----------|------|-------------|
| F1 | N56°E | Vertical | 3,6 |
| F2 | N44°E | NW | 2,1 |
| F3 | N44°E | NE | 2,9 |
| F4 | N56°E | Vertical | 1,3 |
| F5 | N90°E | Vertical | 2,6 |

| | | | |
|------|--------|----------|-----|
| F6 | N60°E | NE | 2,1 |
| F7 | N56°E | Vertical | 2,9 |
| F8 | N56°E | Vertical | 1,6 |
| F9 | N90°E | Vertical | 2,3 |
| F10 | N90°E | Nord | 3,5 |
| F11 | N107°E | NW | 2,6 |
| F12 | N65°E | NW | 3,5 |





| F13 | N65°E | Vertical | 1,5 |
|-----|-------|----------|-----|
| F14 | N70°E | Vertical | 2,5 |
| F15 | N90°E | Nord | 2,3 |
| F16 | N90°E | Nord | 1,2 |
| F17 | N32°E | Vertical | 2,3 |
| F18 | N70°E | Vertical | 2,6 |
| F19 | N90°E | NW | 2,6 |
| F20 | N70°E | Vertical | 3,6 |
| F21 | N80°E | NW | 3,6 |

| F22 | N65°E | Vertical | 1,5 |
|-----|-------|----------|-----|
| F23 | N65°E | Vertical | 2,3 |
| F24 | N47°E | Vertical | 3,6 |
| F25 | N90°E | Vertical | 3,5 |
| F26 | N90°E | Vertical | 1,3 |
| F27 | N110°E | Vertical | 2,3 |
| F28 | N40°E | Vertical | 2,3 |
| F29 | N110°E | Vertical | 2,5 |



### 4.5.3. 2.75D modeling.

**Profile 1**

This profile extends 48.8 km NW-SE through Badzere and Mborguene. It crosses 6 geological formations from NW to SE, namely: porphyroid granite, granite with biotite, Gneiss embrechites, granite of anatexis, schists, biotite and muscovite gneiss (Fig. 2.A). The strongest anomalies are localized in the NW of the profile with an intensity of 177 nT. The basement obtained is made up of granites anataxis which are old magmatic rocks forming the old basement complex and put in place during the first half of the Precambrian. Its maximum depth is h = 3.608 km which agrees with the depths obtained by the Euler convolution (Fig. 5.A). Its susceptibility is S = 0.05 SI. Above, one can observe the embrechite gneisses (S = 0.025 SI), volcano-clastics schists (S = 0.019 SI). This contact between the granito-gneissic rocks and the Lom schists has therefore caused several fractures and faults, represented here by several intrusions:  porphyroid granite (S=0.029 SI), garnet gneiss (S = 0.026 SI), syenites (S=0.035



SI). Our model agrees with previous geological (Poidevin, 1985; Gazel and Gerard, 1954;
Kouske, 2006; Ngako et al., 2003) and geophysical studies (Koch et al., 2012; Owono et al.,
2019). These intrusions were set up during the pan-African orogenesis (Eno Belinga, 1984) and
are present in our geological map (Fig. 1.B).

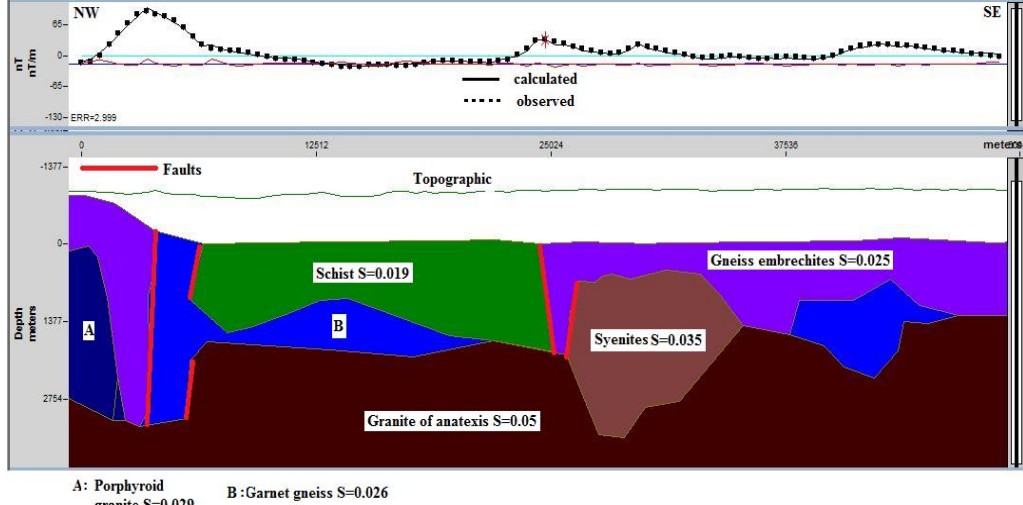

**Figure 5.A** 2.75D model obtained from profile P1.
**Profile 2**
The profile 2 extends 46 km along the NW-SE direction through Bétaré-oya and Kissi. It crosses
5 geological formations: Biotite leptinites gneiss, quartzite with muscovite schists, schists,
biotite and muscovite gneiss, alkaline granite (Fig. 2.A). The lowest anomalies are localized in
the NW of the profile with an intensity of -43.4 nT, while the strongest are on the edge of the
Lom schists with a maximum value of 65.6 nT. The obtain basement is made up of anatexite
granites (S = 0.05 SI), intruded by strongly magnetized rocks such as syenite (S = 0.044 SI),
ryolite (S = 0.037 SI) and anatexic biotites (S = 0.048 SI). Upstream, one can note the
embrechite gneisses (S = 0.025 SI) discordant to volcano-clastics schists (S = 0.023 SI) located
above the metasediments rocks (S = 0.003 SI). One can also observe several intrusions
micaschists (S=0.0186 SI), Graphite (S=0.00012 SI) and Garnet gneiss (S=0.027 SI). The




geological layers obtained are located below the topography and the maximum depth is h =
3.419 km (Fig. 5.B), in agreement with the data resulting from the Euler deconvolution. The
model from this profile is in accordance with previous studies (geology, seismic, magnetic etc.).
We note intrusions from the pan-African orogenesis (Poidevin, 1985; Gazel and Gerard, 1954;
Kouske, 2006; Ngako et al., 2003; Koch et al., 2012; Owono et al., 2019; Eno Belinga, 1984),
located in our geological map (Fig. 1.B).

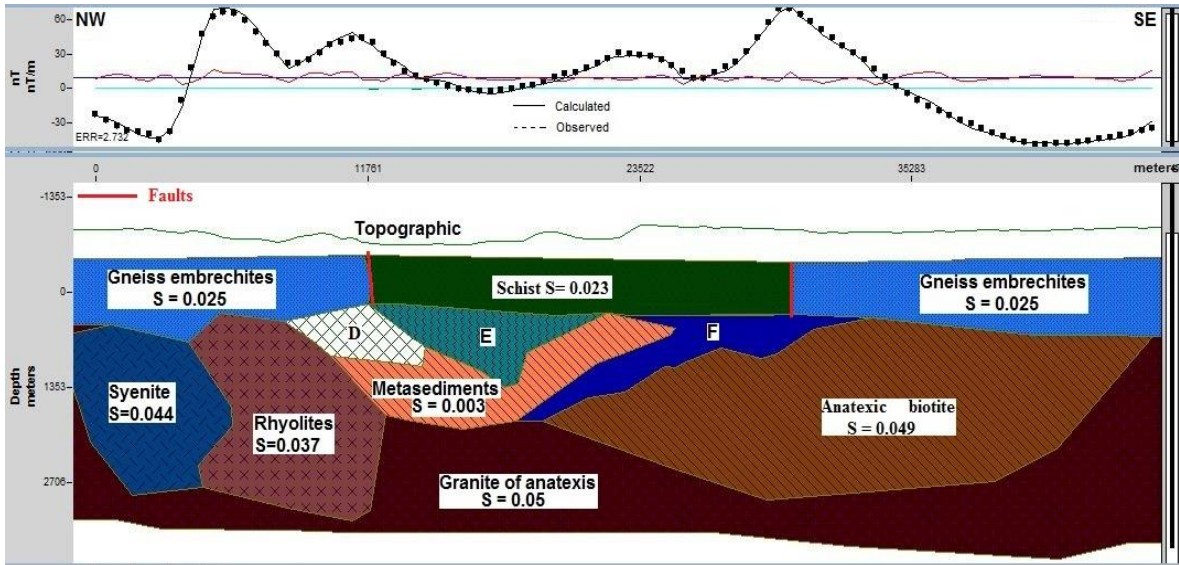

**Intrusive bodies**
**D : Micaschists S=0.0186**
**E : Graphite S=0.00012**
**F : Garnet gneiss S=0.027**

**Figure 5.B**    2.75D model obtained from profile P2.

**5. Regional analysis of the 2.75D models**

The geological synthesis of Cameroon allows us to have a lithostratigraphic sketch of the

Lom Formation. Recently, the near-surface work Mboudou et al., (2017) at Betare-Oya
proposes the lithological model with top soil, saprolites, sandy layer, conglomeritic sand and
schist formations.

On our model from profile 2 that passes through the locality of Bétaré oya, we observe that

the first layers of rocks encountered are well below the topography that is explained by the fact





that the method used allows us to highlight the structuring of deep formations. This would have
the effect of hiding the superficial (sediments) hence the observed shift. Thus, the first
formation detected on our models at Betare-Oya is schist. We can therefore complete this
lithological model with the formations of the pan-African basement highlighted by our
geophysical methods (Fig. 5.C) and propose the litho-stratigraphic model updated below (table
2). Crustal formations in our model are in accordance with those obtained by Jean Benkhelil et
al., (2002) from seismic data south Cameroon and summary above and geological study of
Mboudou et al., (2017).


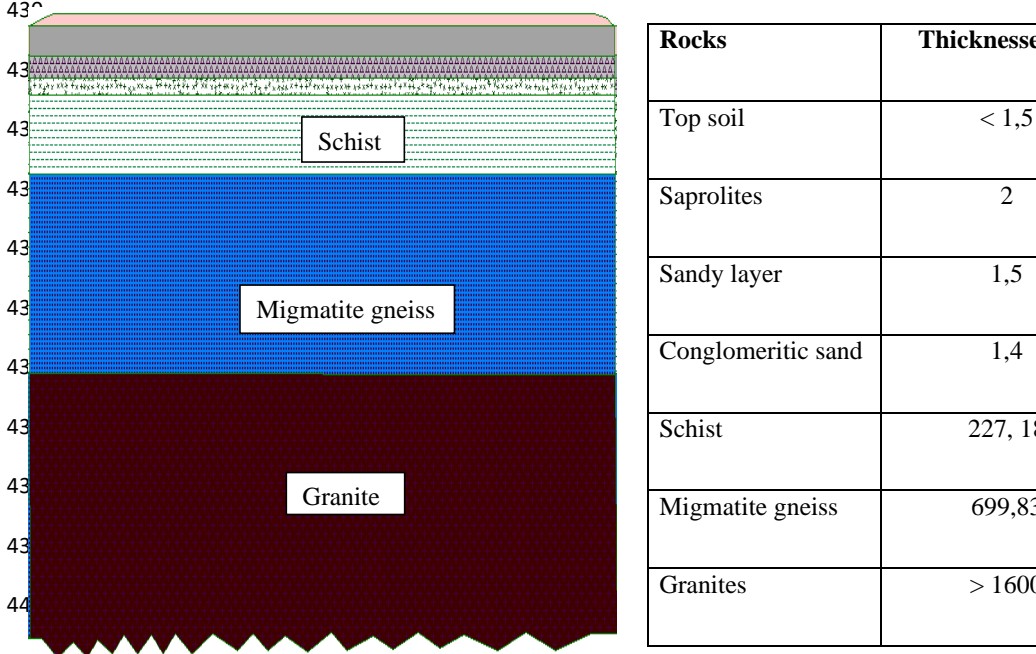

| Rocks | Thicknesses (m) |
|---|---|
| Top soil | < 1,5 |
| Saprolites | 2 |
| Sandy layer | 1,5 |
| Conglomeritic sand | 1,4 |
| Schist | 227, 18 |
| Migmatite gneiss | 699,83 |
| Granites | > 1600 |

**Table 2** Nature of formations.


**Figure 5.C**   Improve sketch of lithologic profile of Betare Oya basin. The scale map is 2/1000.
This model confirms the granite-gneiss nature of the pan-African base.
The major faults highlighted in this work controlled by the Betare - Oya shear zone (BOSZ)
belong in fact to a wider network of faults found on the Pan-African and which would extend
to the São Francisco Craton (SFC) by the central Cameroon shear zone (CCSZ). Indeed, the


work of Toteu et al., (2004) suggests that the Reghane shear zone, which during the whole Pan-
African evolution (650-580 Ma) only recorded dextral wrench movement, can be considered as
a major boundary separating mobile domain in two (Fig. 5.D) - a western part where the
tectonics is controlled by the motion of the WAC and an eastern part controlled by the motion
of the Congo craton.

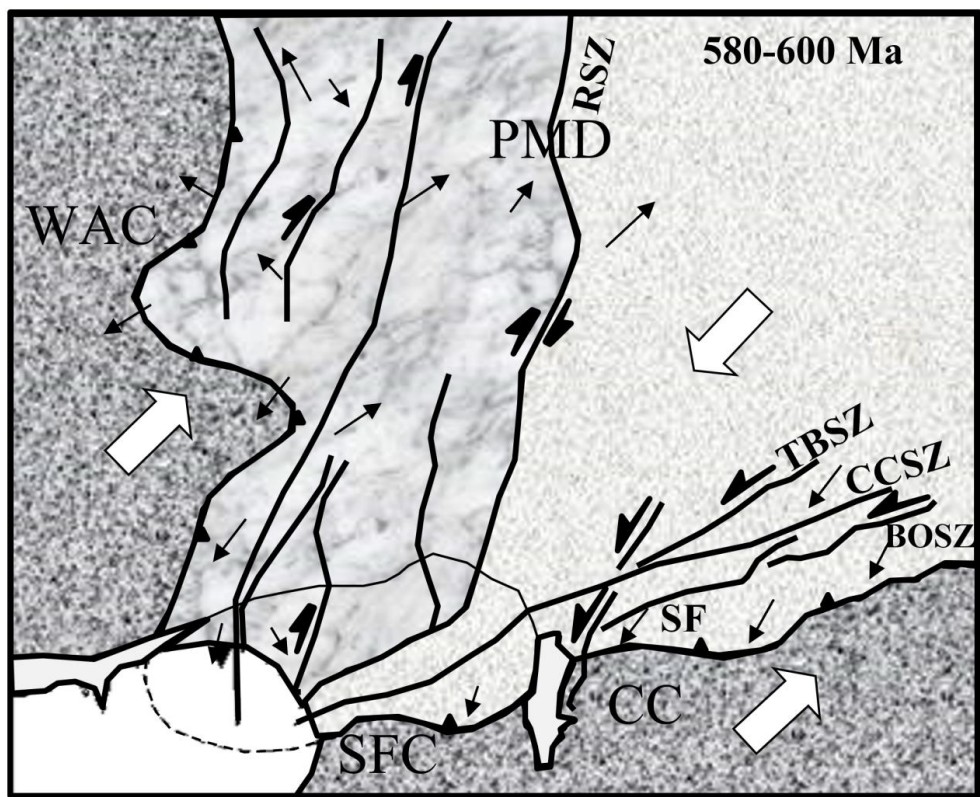


**Figure 5.D**    The Pan-African mobile domain (PMD) between the West Africa craton (WAC)
and the São Francisco (SFC) and Congo (CC) cratons showing two sub-domains, west and east,
separated by the Raghane Shear Zone (R.S.Z.). Horizontal lines represent the Tcholliré‑Banyo
shear zone (T.B.S.Z); central Cameroon shear zone (C.C.S.Z); Sanaga fault (SF); Betare-Oya
shear zone (BOSZ). Small arrows correspond to stretching lineation's and large arrows to
movement directions of blocks during D3 (600‑580 Ma). Toteu et al., (2004) modified (initial
document is available in a public domain).

**6. Discussion**
The structural map obtained (Figure 3.A) shows a great disparity in the distribution of
lineaments which can be explained in part by the general tectonics of the area. Hence, the



collision between the stable Archean craton in the South and one of the two Paleoproterozoic
blocks in the north during the Pan-African orogeny 700 Ma, would have caused a flattening of
the basement and intrusions in the old Precambrian basement, causing the major NE-SW
oriented lineaments related to the Lom schists. According to the Cameroon geological
synthesis, these intrusions are identified as granitic batholiths placed during regional
deformation D1 and D2.
On both sides of the Lom series, there are major NE-SW lineaments representing the
bounding faults of the Lom series with the granite-gneiss rocks. The E-W; NE-SW and N-S
lineaments may represent major tectonic structures marking the change in the structural
direction between the trans-Saharan (N-S) and the Oubanguides (E-W) chains.
At the local scale, the deformation D2 is characterized by L2 lineations representing here
stretches of quartz minerals-oriented E-W. The ENE-WSW oriented lineaments appear to
correlate with the mylonitic deformations occurring during the D3 phase while the ones
trending NW-SE related to senestral and dextral recesses and represent fractures with or without
lode flow. These structures much more abundant near Mararaba and could be the target for
future mining studies.
The geoelectrical study of Nih Fon et al. (2012) in our study area identified NE-SW
oriented irregular anomaly zones. These correlate with the quartz veins known in the region
and are aligned with the regional shear zone. The morphological units identified also present
NW-SE, N-S, NE-SW and E-W directions. In addition, Kouske (2006) reveals that the
hydrographic network of the study area has two major directions, NE-SW and NW-SE and it is
dense and dendritic type.
The P1 and P2 models obtained can be used as pseudo 3D imagery of the Lom basement.
Previous geological studies indicate that the area was a subject to intense metamorphic activity
during Neoproterozoic that has resulted in schist formation (Coyne et al., 2010). The contact
between this schistous series and the gneissic and granitic rocks of the basement resulted in





multiple fractures and faults (Gazel et al., 1954; Soba, 1989). The litho-stratigraphic sketch
proposed by our models derived from the magnetic profiles and work of Mboudou et al., (2017)
are consistent with previous geological work that asserts that the Pan-African basement would
be made up of migmatites and granitic to ortho-gneissic and biotite rich rocks (Poidevin, 1985;
Gazel and Gerard, 1954; Kouske, 2006; Ngako et al., 2003; Koch et al., 2012; Owono et al.,
2019; Eno Belinga, 1984).
From the mining point of view, the artisanal gold indices are places located near the
Lom and Pangar rivers (Nih Fon et al., 2012). These alluviums correlate with NE-SW trends in
our structural map. Since the structures in our study area are structurally guided, it can be
concluded that the alluvial deposits observed and exploited by residents are some signs that
have been leached and transported by the waterways. Overall, the geological structures obtained
from the data processing correspond to the ductile-brittle structures such as shear zone and
faults. These structures constitute pathway for both mineralizing fluids and ground water. Since
several gold mines exist in Betare-Oya area, the new mapping approach could be an important
guide for the identification of the structures that control the gold mineralization in the area.
**7. Conclusion**
In this work, some new analysis techniques were applied on aeromagnetic data to delineate the
sub-surface structures. The results obtained highlight the axes of compression, folding and
shearing; mylonitic veins (veins are at the outcrop's scale) several kilometers long and oriented
NE-SW. The regional and local structural settings of the area are characterized by major faults
and other structural elements mainly striking in the NE-SW, NW-SE, ENE-WSW, N-Sand E-
W directions. Major trend in the NE-SW direction represents the dominant tectonic trend which
is the prolongation of the Central Cameroon Shear Zone (CCSZ) in the study area. Several folds
and faults evidenced by this study correlate with past studies while others are inferences. The
depths of major accidents in the area have been estimated between 1.2 to 3.6 km and the NE-
SW structures on our structural map are proposed here for a possible gold exploration. The





models from the P1 and P2 profiles have enabled: to propose a structuration of the superficial
crust of the Lom highlight the main rocks and intrusions responsible of the observed anomalies
(porphyroid granite, garnet gneiss, syenites, micaschists, Graphite and Garnet gneiss), identify
deep and shallow fractures, their depths and to propose a lithostratigraphic model in agreement
with the previous works. Finally, we note that the tilt angle coupled to the upward continuation
is an interesting tool for 3.D modeling.
**Data Availability**
The data used to support the findings of this study are available from the corresponding
author upon request.
**Author Contribution**
Christian Emile Nyaban performed the data analyses, modelling and preliminary interpretation
including preparation of the manuscript in conjunction with all the co-authors; Theophile
Ndougsa-Mbarga design the topic, gives the orientations for the investigation and reviewed the
quality of the models and related interpretation and the entire manuscript **;** Marcelin Bikoro-
Bi-Alou defines the criteria and the physical parameters for the 2D3/4  modelling with the first
author; Stella Amina Manekeng-Tadjouteu and  Stephane Patrick Assembe have worked on the
review of quality and quantitative  analyses of respectively maps and 2D3/4 models.
**Competing Interest**
The authors declare that there are no conflicts of interest regarding the publication of this paper.
**Acknowledgements**
The authors thank the reviewers for their valuable comments.

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
