# Peer review of "Multi-scale analysis and Modelling of aeromagnetic data over the"

_Solid Earth, 2020_

## Referee Comment (RC1) · Bruno Gavazzi (Referee) · 24 Sep 2020

General comments The article aims to provide new insights on structural features of the Bétaré-Oya area in Cameroon through the use of different potential field transformation and modeling from aeromagnetic data. I am not very familiar with the geological context of the area so I will focus my review on the methodological aspects and on the interpretation of the magnetic data. I find the approach interesting and interpretations are mostly coherent. Unfortunately the methodology is not well described and some assumptions are not explained enough. The paper would need a strong revision on

this aspect to provide a solid base for the discussion. In addition, some references are missing in the reference list. In its present state, it is hard to evaluate the validity of the discussion.

My advice would be to review the literature on aeromagnetic methods starting with Nabighian et al (2005), The historical development of the magnetic method in exploration, Geophysics 70 (6). 33ND-61ND https://library.seg.org/doi/10.1190/1.2133784

I provide more details in the specific comments section.

In the future, it would be useful to provide the DOI of the references, it is easier to find them that way.

Specific comments L 36 – 39 You should provide an explanation on why you want to achieve that in your particular case. It reads as if removal of large wavelength is always done on aeromagnetic dataset, but it depends on the application. Moreover, Ndougsa et al 2007 is about gravity and not magnetic data. L 39 – 40 Here it also reads as if in the general case shallow bodies are associated to iron deposits. It obviously depends on the context, I think you should make an argument for your special case and context. L 47 - 49 Verduczco et al 2004 did not develop tilt derivative but discuss its use, please read the suggested reading provided at the end of the paper to correct your statements. Also the tilt is used as an edge detector for vertical contacts, not for all shapes. L 50 – 53 I think you should say why you do not use the method from Salem et al 2008 which seem well suited to what you want to do, is it because of the use of second order derivatives and your signal/noise ratio? L 142 what does it mean a sensitivity of 0,5 nT? Is it only the sensor? What about the FOM? What is the overall precision (differences at the crossing points?). Also, I could not find the reference, is it an internal report? If so could it be published on an open archive in order to be available? L 144 is there any special reason for a grid step of 850 m? (it is common to use half the profile spacing or the profile spacing) L144 – 145 Is it relevant to precise that the digitization was well done? L146 why do you use IGRF 1984 and not the latest available? (also you should

cite the associated publication) L 150 – 153 Blakely describes the upward continuation, not how to remove the regional effect, also the upward continuation was proposed by Henderson and Zietz (1949) https://library.seg.org/doi/10.1190/1.1437560. Please be more precise L 163 – 164 this works for vertical contacts, how do you deal with non vertical edges? Or what is your argument for an assumption of only vertical contacts? L 166 – 169 I do not understand how coupling upward continuation and tilt do what you say, the advantage of tilt is that it is not dependent of depth of sources. Could you explain better? L 198 – how does reduction to equator gives position? Why don't you use reduction to the pole (there are many techniques to deal with proximity to equator, please see Nabighian 2005), how can you assume only induced magnetization? (you cannot reduce to the pole/equator with remanent magnetization) L 203 – 221 "positive" or "negative" anomaly has no sense, a magnetic anomaly has always a positive and a negative parts. Also what do you mean by bipolar? Is it dipole? You should reformulate this section to make it more scientifically correct. L 241 – 249 Why don't you use the IGRF as regional field? Zeng (1989) is not in the reference list. I had never seen this method, could you provide references and/or an explanation on why you choose this technique? L 322-326 It works only for vertical contacts L 524 -525 "data available upon request" is not an open science statement. Could you upload the data on an open archive (such as zenodo) or are they confidential?

Technical corrections You use sometimes modelling (British english), and sometimes modeling (US). Please choose one. L 43 – Oruç et al is not in the reference list L 44 "In the last few year" and then you cite literature from 1985. Would be more accurate to reformulate that. Fig 1B I cannot see well the faults as indicated in the legend. Also, what are "tectonic lines" L 149 I would remove "theory" as you do not discuss the theory behind it L 150 the first sentence is not understandable

Please also note the supplement to this comment:
https://se.copernicus.org/preprints/se-2020-111/se-2020-111-RC1-supplement.pdf

[Figure]

Interactive
comment

---

## Referee Comment (RC2) · Anonymous Referee #2 · 2 Nov 2020

In this manuscript, the aeromagnetic data over the Bétaré-Oya area in the Eastern Cameroon were mainly used to derive geological interpretations associated with local tectonic structures in this area. By means of several techniques, including tilt derivative, Euler deconvolution, upward continuation and the 2.75 modelling, fine magnetic maps were presented, related to the tectonic lines and faults. The results and conclusions from this manuscript were well described and fruitful. However, readers may be still interested in several additional details before this manuscript published: 1. As the only observed magnetic dataset used in this study, the original locations and intensities of the aeromagnetic data should be shown. 2. What are the new findings in this manuscript compared to previous studies? Is it that in this work the sub-surface tectonic structures were for the first time related to the magnetic data? 3. How to evaluate the error and role of the 2.75D modelling to obtain new regional results in this study?
* * *

---

## Author Comment (AC1) · 1 Dec 2020

Responses to reviewer 1 comments on the paper titled: Multi-scale analysis and Modeling of aeromagnetic data over the Bétaré-Oya area in the Eastern Cameroon, for structural evidences investigations.

Dear Chief Editor, In general, all the reviewer remarks, and recommendations have been taken into consideration. The authors make changes and suggestions in yellow in the MS text, but in blue below responses are given to all the remarks. The authors

are indebted to him for his valuable remarks. Reviewer1: Bruno Gavazzi

I- Some references are missing in the reference list L 43 – Oruç et al is not in the reference list L 241 – 249 Zeng (1989) is not in the reference list. We added the references in the list. 1. L 552 – 554 (Oruç et al., 2011) 2. L 645 – 646 (Zeng, 1989)

II- The methodology is not well described 1. L 47 - 49 Verduczco et al 2004 did not develop tilt derivative but discuss its use, please read the suggested reading provided at the end of the paper to correct your statements. Also, the tilt is used as an edge detector for vertical contacts, not for all shapes. L 47 - 49 We have changed accordingly with the recommendation.

5. L 142 a)- what does it mean a sensitivity of 0,5 nT? Is it only the sensor? L 142 b)- What about the FOM? L142 c)- What is the overall precision (differences at the crossing points?): It is the recording sensitivity of the magnetometer used and considered as the noise level. The figure of merit (FOM) in the present data has been considered as being the same as the noise level and their combination accepted as the overall precision. L 142 d)- Also, I could not find the reference, is it an internal report? If so could it be published on an open archive in order to be available? : It's an internal report which is available in the format of CD rom at the " centre of geological & mining information (CIGM)" in the Ministry of mines but not online, and the corresponding author has a copy of this CD rom.

9. L 150 – 153 Blakely describes the upward continuation, not how to remove the regional effect, also the upward continuation was proposed by Henderson and Zietz (1949) https://library.seg.org/doi/10.1190/1.1437560. Please be more precise: L 152 – 153 Done.

12. L 198 – how does reduction to equator gives position? Why don't you use reduction to the pole (there are many techniques to deal with proximity to equator, please see Nabighian 2005), how can you assume only induced magnetization? (you cannot reduce to the pole/equator with remanent magnetization): At low magnetic latitudes

(between -15 ° and 15 °) as is the case here, the N-S magnetic field directions are amplified by the reduction at the pole and there is a great risk to have an exaggerated noise by amplifying pre-existing one. In this case the map reduced to the pole is un-readable and unstable. To overcome this problem, it is preferable to apply the reduction to the equator. In theory, the reduction to the equator transforms an anomaly caused by a magnetized body having a non-zero inclination, into another anomaly that would be associated with the same body if the inclination of the magnetization were zero. From a spatial representation point of view, the anomaly changes from any shape to a characteristic symmetrical shape, with a latitudinal central lobe framed on the north and south by two lobes of opposite sign to the first. For a given anomaly, the shape of the anomaly reduced to the equator obtained therefore makes it possible to judge pos-teriori the parameters of inclination and declination of the magnetization. If the shape obtained is the most symmetrical as possible, this means that these starting parame-ters are close to actual parameters of the magnetization. Moreover, if these starting parameters are close to those from the current magnetic field, we can then hypothe-size an induced behavior of the magnetic body, or conversely, its essentially remanent character. The disadvantage of pole reduction (Baranov, 1957; Baranov and Naudy 1964) is that it requires knowledge of the direction of the source magnetization vector which is often a difficult parameter to know. This is why it is commonly assumed that the magnetization of the source is purely induced, consequently it has a direction iden-tical to the direction of the magnetic field assumed to be known, for example the global geomagnetic models (eg IGRF International Geomagnetic Reference Field) (Feumoe et al., 2012).

13. L 203 – 221 "positive" or "negative" anomaly has no sense, a magnetic anomaly has always a positive and a negative part. Also what do you mean by bipolar? Is it dipole? You should reformulate this section to make it more scientifically correct. L 215 – 216 The reformulation is done

III- Some assumptions are not explained enough 1. L 36 – 39 You should provide

an explanation on why you want to achieve that in your particular case. It reads as if removal of large wavelength is always done on aeromagnetic dataset, but it depends on the application. Moreover, Ndougsa et al 2007 is about gravity and not magnetic data: L 37 – 38 There was a confusion on the reference. The good one is Ndougsa et al., 2013. We revised the text and have followed the recommendation.

2. L 39 – 40 Here it also reads as if in the general case shallow bodies are associated to iron deposits. It obviously depends on the context; I think you should make an argument for your special case and context: L 39 – 41 We revised the text and have restricted this part to our magnetic case.

4. L 50– 53 I think you should say why you do not use the method from Salem et al 2008 which seem well suited to what you want to do, is it because of the use of second order derivatives and your signal/noise ratio?: We have used Salem approach for the location of vertical contacts and source depth. In addition, because the identified source has a non-uniform volume from the roof to the bottom, we examine how this volume varies with depth by using upward continuation of magnetic anomaly.

6. L 144 is there any special reason for a grid step of 850 m? (it is common to use half the profile spacing or the profile spacing): There is no special reason, it was an error of transcription by the co-author who was in charge of the edition of the manuscript, we use effectively 750 m, thanks for that.

7. L144 – 145 Is it relevant to precise that the digitization was well done? :We did not see on the MS where this sentence "the digitization was well done" is mentioned.

8. L146 why do you use IGRF 1984 and not the latest available? (also, you should cite the associated publication): It is a mistake. We used IGRF -70 which is the nearest (Reeves, 2005)

10. L 163 – 164 this works for vertical contacts; how do you deal with non-vertical edges? Or what is your argument for an assumption of only vertical contacts?: It is

certain that a great part of the work is devoted to the identification of vertical contacts, but we used also Euler deconvolution to do the inventory on non-vertical edges. The focus was to delineate the structural infrastructure of the near surface of this area under study, which has many small-scale mechanized gold exploitations. These vertical contacts at the near surface could be preferred zones of gold bearing.

11. L 166 – 169 I do not understand how coupling upward continuation and tilt do what you say, the advantage of tilt is that it is not dependent of depth of sources. Could you explain better?: Salem et al.,(2007) proposed the use of tilt angle for the localization of vertical contacts. Knowing that the upward continuation operator can attenuate short wavelengths and allow to visualize long wavelengths (Henderson and Zietz, 1949), We can therefore use it for a better visualization of the behavior of contacts with depth. Thus, we have: - Generated the TMI maps reduced to the equator and then apply upward continuation for 1 and 2 km; - Generated the vertical contacts of these different three maps using Salem et al. (2007; 2008); - superimposed finally the different contact maps obtained to evaluate the continuity of the sources. By applying those principles, it is generally observed from the obtained maps that: i- They are not identical, which could mean that the contacts situated at the near surface could be masked by those located at the subsurface or in depth; ii- There are some vertical contacts that narrowed with depth. This could be interpreted as a sign of crustal thinning of the source of the anomaly with depth; iii- In some places, a lateral displacement of the contact is identified. It could suggest here, a dip of the source in the concerned direction.

14. L 241 – 249 Why don't you use the IGRF as regional field? I had never seen this method, could you provide references and/or an explanation on why you choose this technique?: We subtracted the IGRF from the TMI, to obtain the crustal field. Considering that, we want to conduct a near surface investigation for a possible mineral prospecting because our study area is the object of semi-mechanized artisanal gold mining, we generated an optimal regional using the approaches of Zeng et al. (1989), Marcel Jean et al. (2016). The approach consists in generating the maxima of the

extended maps at different altitudes, then extracting a database of these points that we can then compile in the Excell software. The valid altitude for the regional map to be retained will be the one from which the curve tends towards zero.

15. L 322-326 It works only for vertical contacts: Please see clarifications given in the response for L 163-164.

16. L 524 -525 "data available upon request" is not an open science statement. Could you upload the data on an open archive (such as zenodo) or are they confidential?: The data belongs to the Project of Capacity Building for the Mining Sector (PRECASEM) of Cameroon and this project is under the Minister of Mines, Industry and Technological Development.

IV- Technical corrections

17. You use sometimes modelling (British english), and sometimes modeling (US). Please choose one: Choice is done by using US.

18. L 44 "In the last few year" and then you cite literature from 1985. Would be more accurate to reformulate that. Fig 1B I cannot see well the faults as indicated in the legend. Also, what are "tectonic lines": L 44 all the recommendations are done. tectonic lines are red lines on the map 19. L 149 I would remove "theory" as you do not discuss the theory behind it:L 149 Done.

19. L 150 the first sentence is not understandable: L 150 – 151 Reformulations done

We humbly hope that the clarifications and the corrections made after receiving the reviewer 1 remarks & recommendations are satisfactory. Your kind reaction is awaited.

Sincerely yours

Please also note the supplement to this comment:
https://se.copernicus.org/preprints/se-2020-111/se-2020-111-AC1-supplement.pdf

[Figure]

[Figure]

[Figure]

**Fig. 1.** Figure 1.B Geological map of the study area (Gazel and Gerard, 1954 modified as a document available in a public domain). In the center we have the Lom series marked by its greenschist facies. We c

---

## Author Comment (AC2) · 1 Dec 2020

Responses to reviewer2 comments on the paper titled: Multi-scale analysis and Modeling of aeromagnetic data over the Bétaré-Oya area in the Eastern Cameroon, for structural evidence investigations.

Dear Chief Editor, In general, all the reviewer 2 remarks, and recommendations have been taken into consideration. The authors make changes and suggestion in yellow in the MS text, but in blue below responses are given to all the remarks. The authors are

indebted to him/her for his/her valuable remarks. Reviewer2: Anonymous

1. As the only observed magnetic dataset used in this study, the original locations and intensities of the aeromagnetic data should be shown: The original data set belongs to the Ministry of Mines, Industry & Technological Development. We got the TMI maps for our use, with a condition of not sharing with a third party. We digitized them and then, got all the results that are in the present manuscript submitted for publication. We are sorry not to be able to give:

2. What are the new ïñandings in this manuscript compared to previous studies? Is it that in this work the sub-surface tectonic structures were for the ïñrst time related to the magnetic data?: The subsurface tectonic structures are for the first time related to the magnetic data, but we can mention the fact that, some other geophysical methods are used in investigating neighbouring areas respectively in the southern part (Pepogo et al., 2018, using audiomagnetotellurics soundings, Tadjou et al., 2009 by modelling and interpreting gravity data) and the southern East part ( Owono-Amougou et al., 2019). All these previous studies are cited in the sub-section 2.3. Our results are corelated to those from other geophysical studies realized in surrounding sites by using the methods cited above. Here they are: i)-Several major families of faults were mapped. Their orientations are ENE-WSW, E-W, NW-SE, N-S with a NE-SW prevalence. The latter are predominantly sub-vertical with NW and SW dips and appear to be prospective for the future mining investigation. ii)-The evidence of compression, folding and shearing axis, was concluded from superposition of null contours of the tilt-derivative and Euler deconvolution. The evidence of the local tectonics principally due to several deformation episodes (D1, D2 and D4) associated with NE-SW, E-W, and NW-SE events, respectively. iii)- Depths of interpreted faults ranges from 1000 to 3400 m. iv)-Several linear structures correlating with known mylonitic veins were identified. These are associated with the Lom faults and represent the contacts between the Lom series and the granito-gneissic rocks; we concluded the intense folding caused by senestral and dextral NE-SW and NW-SE stumps. v)- We propose a structural model of the top

of the crust (schists, gneisses, granites) that delineates principal intrusions (porphyroid granite, garnet gneiss, syenites, micaschists, Graphite and Garnet gneiss) responsible for the observed anomalies. The 2.75D modeling revealed; many faults with a depth greater than 1200 m and confirmed the observations from RTE-TMI, Tilt derivative and Euler deconvolution. vi)- We developed lithologic profile of Betare Oya basin.

3. How to evaluate the error and role of the 2.75D modelling to obtain new regional results in this study?: The error on our 2.75 D model is evaluated by minimizing the difference between the measured value and the theoretical curve automatically generated by the GM-SYS operator. The smaller this difference, the ac-ceptable are the values obtained. The role of 2.75 D modeling is to bring out an imagery of the different geological layers in the subsoil, responsible of the magnetic responses obtained at the surface

I humbly hope that the clarifications on the corrections made after the reviewer 2 remarks & recommendations are satisfactory. Your kind reaction is awaited.

Sincerely yours

Please also note the supplement to this comment:
https://se.copernicus.org/preprints/se-2020-111/se-2020-111-AC2-supplement.pdf

---

## Author Response (AR2)

**Responses to reviewer Bruno Gavazzi comments on the paper titled**: **Multi-scale analysis and Modeling of aeromagnetic data over the Bétaré-Oya area in the Eastern Cameroon, for structural evidence investigations.**

Dear Chief Editor,

In general, all the reviewer (cited above) remarks, and recommendations have been taken into consideration. The authors make changes and suggestions in yellow in the MS text, but in blue below responses are given to all the remarks. The authors are indebted to him for his valuable remarks.

**General comment and remarks**

1. The revised manuscript that was strongly improved from the first version. Corrections suggested by the reviewers have been nicely implemented and the methodology part is now more detailed.

   a)- Nonetheless, I think the paper needs some precisions on the tilt method before publication: I understand you use the tilt depth method on data reduced to the equator (RTE), but Salem et al 2007 described the tilt derivative on data reduced to the pole (RTP) and stated that this necessity is one of the limits of their method. You should explain how it works with your RTE data and what the limits of such an approach are compared to tilt on RTP data (and why you can use it for your case). I deduced that you can consider that the shallow contacts are vertical and that therefore you can use tilt to map them. But I think it is not stated clearly enough: please make it clear that you have proofs or strong hypothesis that the contacts are vertical before mapping them: The RTE operator acts like the RTP's one, that means, it allows the tilt of the magnetization to be removed. This thus brings the anomaly of the magnetic field in line with the causative source (Salem et al., 2012; Feumoe et al., 2012). But, in the case of low latitude areas between -15° and 15° as is the case for our study area, the N-S directions are amplified by the reduction at the pole and there is a higher risk of an exaggerated reinforcement by amplifying a pre-existing noise. When applying the RTP operator in low latitude areas, as in our case, the map obtained is instable, a little fuzzy, and more gives anomalies values that do not correlate with those from the aeromagnetic maps of the adjacent areas (Feumoe et al.,2012, Owono-Amougou et al., 2019 & 2020) where the RTE was applied. It is necessary to mention that, in the case of the adjacent areas situated just below (Feumoe et al.,2012, Owono-Amougou et al., 2019 & 2020) our study area, the RTP was tested by the concerned authors, their maps were fuzzier, and the noise is highly amplified. To solve that problem in low latitude regions, it is better to apply the reduction to the equator (Feumoe et al., 2012, Owono-Amougou et al., 2019 & 2020).

Concerning our hypothesis for the vertical contacts, we used the results from surroundings areas where the RTE was applied and we combined those results based on the same hypothesis with geological facts derived from field observations (Soba, 1989) covering our study area. Concerning the limits of the tilt derivative on data RTE, we do the constatations cited below:

- With the RTE, we used the hypothesis of a magnetic induced anomaly, which has the same direction as the geomagnetic field, given by the global geomagnetic field models (IGRF), and we ignore/neglect its remanent part in the rock.

- To have satisfactory results using the tilt angle on magnetic data analysis and modelling, especially in the determination of the vertical contacts in low latitude areas, the RTE operator becomes a mandatory constraint.

In our case, investigating a low latitude area we engaged the:

- Use of the Euler deconvolution to identify various potential contacts including the vertical one and then compare them with those derived from the tilt results.

- Care to correlate our results with the previous geological investigations over and outside on one hand and geophysical studies over adjacent areas particularly the aeromagnetic investigations on the second hand.

**Technical corrections and suggestions**

2. I also provide hereafter some technical corrections and suggestions:

L 39: These: Done.

L 118: "were" instead of "are"?: Done.

L134 to 136: is is plural (and should be "...maps have been…") or a singular map ( and it should be "The aeromagnetic...": Done.

L142: I think you should replace "fields" (general case) with "magnetic field") your case: Done.

L145: "their effect" instead of "regional effect"? The regional effect might be defined as something different, using their (the deep sources) avoids the confusion : Done.

L155: I would add "vertical" before "contacts" (because for non-vertical contact h=+/-90): Done.

L168: it could be understood that the operator highlights only the vertical contacts (it is not true, but for the other the position depends on the slope angle), I could propose: "Computed the position of the contacts considered as vertical...": Done.

L 197: it reads as if the operator has been done in 1970, could you rephrase that it is clear that it was performed with the inclination and declination of 1970, the ref to the latest IGRF would be appropriate here (IGRF-12, Thébault et al 2015):
a)-Reformulations done.
b)- Our choice which is based on the use of the corresponding IGRF, by inserting into Oasis Montaj 8.4, the data collection date (1st dd/mm/yy), has led to declination and inclination values for the IGRF, which is included in the interval of validity defines by the latest IGRF-12 (Thébault et al.,2015).

L216: "magnetization" (remanent+induced) instead of "susceptibilities" (only induced) (strictly

speaking it is even contrast of magnetization of the causative bodies with the

neighboring/surrounding materials): Done.

We humbly hope that the clarifications and the corrections made after receiving the reviewer remarks & recommendations are satisfactory.

Your kind reaction is awaited.

Sincerely yours